# Bending Analysis of Polymer-Based Flexible Antennas for Wearable, General IoT Applications: A Review

**DOI:** 10.3390/polym13030357

**Published:** 2021-01-22

**Authors:** Muhammad Usman Ali Khan, Raad Raad, Faisel Tubbal, Panagiotis Ioannis Theoharis, Sining Liu, Javad Foroughi

**Affiliations:** 1School of Electrical Computer and Telecommunication Engineering, University of Wollongong, Wollongong, NSW 2522, Australia; muak803@uowmail.edu.au (M.U.A.K.); raad@uow.edu.au (R.R.); faisel@uow.edu.au (F.T.); pit289@uowmail.edu.au (P.I.T.); sl527@uowmail.edu.au (S.L.); 2Westgerman Heart and Vascular Center, University of Duisburg-Essen, 45122 Essen, Germany

**Keywords:** polymers substrates, flexible electronics, flexible antennas, internet of things (IoTs), wearable applications

## Abstract

Flexible substrates have become essential in order to provide increased flexibility in wearable sensors, including polymers, plastic, paper, textiles and fabrics. This study is to comprehensively summarize the bending capabilities of flexible polymer substrate for general Internet of Things (IoTs) applications. The basic premise is to investigate the flexibility and bending ability of polymer materials as well as their tendency to withstand deformation. We start by providing a chronological order of flexible materials which have been used during the last few decades. In the future, the IoT is expected to support a diverse set of technologies to enable new applications through wireless connectivity. For wearable IoTs, flexibility and bending capabilities of materials are required. This paper provides an overview of some abundantly used polymer substrates and compares their physical, electrical and mechanical properties. It also studies the bending effects on the radiation performance of antenna designs that use polymer substrates. Moreover, we explore a selection of flexible materials for flexible antennas in IoT applications, namely Polyimides (PI), Polyethylene Terephthalate (PET), Polydimethylsiloxane (PDMS), Polytetrafluoroethylene (PTFE), Rogers RT/Duroid and Liquid Crystal Polymer (LCP). The study includes a complete analysis of bending and folding effects on the radiation characteristics such as S-parameters, resonant frequency deviation and the impedance mismatch with feedline of the flexible polymer substrate microstrip antennas. These flexible polymer substrates are useful for future wearable devices and general IoT applications.

## 1. Introduction

Over the last few decades, Flexible Electronic System (FES,) known as flex circuits, have been growing rapidly in industries and organisations such as medical healthcare, energy and power, aerospace, industrial automation, military and defence, sports and entertainment are now the array of various devices are generally comprised of organic substances as a substrate. These substrates create flexible devices which are not only usually characterized by their flexibility, but also their lightweightedness, durability and energy efficiency and are, resultantly, becoming recognized for their huge relevance to healthcare and medical products as well as for defence and wearable electronics. According to a market survey by Research Nester, published in “Global Flexible Electronic Market Overview”, the flexible electronic market annual growth rate is anticipated to expand 19.7% over the period 2017–2024 [1] and is estimated to reach over 30 billion USD in 2028 [2]. For certain electronic and communication applications, the flexibility of different materials is of great importance with extensive use demonstrated in flexible displays, smart tags, wearable products and flexible antennas [3,4]. Accordingly, researchers have incorporated a variety of materials for providing improved flexibility in electronic systems, including the application of polymers, plastics, paper, textiles and fabrics as substrates of these systems. Each of these materials has its own characteristics in terms of how efficiently they can be safely bent, twisted and/or crumpled.

Flexible electronics have become essential for applications requiring flexible displays and biomedical applications with complex curvilinear structures [5,6]. Correspondingly, researchers have experimented on many materials to provide increased flexibility in electronic systems, including polymers, plastic, paper, textiles and fabrics as a substrate of these systems. Each of these materials has its own individual characteristics in terms of how efficiently they can be bent, twisted and/or crumpled [7]. The bendability and flexibility characteristics of these materials make them advantageous for incorporation in designs for future smart electronics systems including application in the Internet of Things (IoTs).

For certain electronic and communication applications, the flexible characteristic of different materials is of great importance, with extensive use spreading into flexible displays, smart tags and wearable products as well as the flexible antenna developments [3,4]. Indeed, the flexible displays and antenna systems are now considered an essential part of personal communication, industry, military, and telemedicine. These flexible devices have many utilizations in health monitoring systems, aeronautics and RFID tagging applications [8,9,10,11,12,13,14,15,16]. Flexible circuits, such as carbon-nanotube thin films on plastic substrates, provide a conformal and lightweight construction. These flexible integrated circuits have many potential areas of application in embedded systems and other areas of electronics [17] with various types of flexible RFID tags are already in widespread use [18,19]. Recent examples of flexible electronics include stretchable organic solar cells which can be used as biological sensors, active-matrix displays and stretchable power sources [20]. Some other novel applications more recently on the market include flexible displays and touch screens [21], electronic paper [22] and skin-like sensing robotic systems [4,23,24], just to name a few.

While flexible electronic devices are constructed with flexible materials such as polymers, plastic, laminates, conductive foils and fabrics, their systems, the FES, can be categorized into four main constructive parts: substrate, backplane, front panel and encapsulation (see Figure 1) which illustrates these four major parts of the FES. The appropriate selection of the first part “substrate”, the base material upon which the whole circuit is produced, is critical. The second part is the backplane, which is a Printed Circuit Board (PCB) with slots for connecting electronic components. The third part, the front panel, is customarily a metal sheet that supports the components and allows certain alteration to system components. An encapsulation layer, which encloses circuitry with a protective covering, is the final component. All of the FES parts must have some degree of bending capability without which it would conflict with the normal function of the FES.

Presently, FES has become crucial for the progressive development in wearable devices which generally include flexible antennas, smart tags and sensors. The FES system covers various fields which depend on its specific applications, such as the development of Printed Circuit Boards (PCB), flexible displays, energy storage and generation, devices applicable for Wireless Body Area Networks (WBAN), see Figure 2. The flexible PCBs, displays and energy storage devices are used extensively in healthcare, entertainment, business, military and space applications.

The Wireless Body Area Network (WBAN) provides communications between body-worn devices and devices in the surroundings. These wearable devices include smart tags such as Radio Frequency Identification (RFID) tags and flexible wearable antennas. The emergence of the WBAN is coming out to combine FES and body-worn devices that can easily be mounted on the human body to allow humans to wear antennas and smart tags instead of carrying them. Research on the development of flexible wearable devices such as antennas and smart tags on a flexible substrate is, therefore, a fascinating area in need of further investigation.

Presently, with the exponential development of these wearable sensors and devices and the high demand of the flexible communication systems, various new challenges have arisen because of the unconventional performance requirements. The typical example, and the relevance to this paper, includes wearable electronics with flexible antennas [25], so it is very important for the flexible antennas to be lightweight, small, durable, moist and heat resistant and most importantly highly flexible without distorting radiation characteristics. For this reason, this investigation focuses on the flexibility and seamless integrity of the flexible antennas.

Traditional antennas are customarily made of conductive wires or by etching metal patterns on rigid substrates. When subjected to stretching or are folded or twisted, these types of antennas become permanently deformed, if not broken, which renders them incompatible for applications that require high bendability and are subject to continuous deformation. Flexible antennas have gained much attention in recent years with their advantage of directly addressing this problem with their high flexibility, but also for their convenient integration with other microwave components [26], lightweightedness, energy efficiency, reduced fabrication complexity, easy mount-ability on conformal surfaces, their low cost and for the abundant availability in the form of substrate films [27]. The concept of flexible wearable antennas has emerged from the progressive evolution of the Flexible Electronic System (FES).

To achieve the aforementioned characteristics for flexible antennas, conventional conductors and substrate materials such as metals and ceramics are not essentially appropriate. This is because these materials are usually rigid, costly, and lack flexibility and mechanical resilience. A lot of research has already explored numerous materials which exhibit suitable properties as a substrate for conductive materials for antennas are conductive polymers [28,29,30,31,32,33,34], conductive threads [35,36,37] and conductive textile [38,39,40,41,42,43,44,45,46,47,48,49,50,51,52]. For dielectric materials, Polyimide (PI) [7,11,26,29,53,54,55,56,57,58,59], Polyethylene Terephthalate (PET) [60,61,62,63,64,65,66,67], Polydimethylsiloxane (PDMS) [68,69,70,71,72,73,74,75,76], Polytetrafluoroethylene (PTFE) [15,77,78,79,80], Liquid Crystal polymers (LCP) [10,81,82,83,84,85,86,87,88] have been explored.

The main objectives and the contributions of this paper relate to recent literature in the field and are subsequently summarised as follows:An overview of the history and chronological advancement in flexible electronics during the last six decades is presented. This illuminates the general structure of the FES, general properties and selection of flexible material to provide the desired flexibility of dielectric substrates for specific applications.A general transmission line model of a flexible polymer-based antenna, impact on radiation characteristics by bending, general properties and the selection of flexible material to provide the desired flexibility of dielectric substrates for specific applications.The most efficient and relevant polymer substrates, in terms of bending and flexibility which is a key challenge for flexible IoT and wearable applications, are elaborated with certain profundity in an overview of related research work in the most recent literature.A detailed comparative analysis of physical, electrical, thermal and chemical properties of flexible polymer materials which have been used as a substrate to provide flexibility during the last few decades, is presented.The bending effects of flexible polymer substrate antennas on radiation characteristics for different frequency requirements are analysed and discussed with reference to the previously published articles.

To the best of the author’s knowledge, this paper is the first comprehensive review paper that provides bending capabilities of the most abundantly-used polymer substrate for flexible antennas. The rest of the paper is organized as follows: Section 2 illustrates the chronological advancement in the field of flexible electronics and the history of flexible substrate materials development. General flexible antennas for IoTs or wearable applications are presented in Section 3. This incorporates a presentation of the basic structure of smart flexible antennas, transmission line model in terms of the impact of bending, wearable electronic devices and bending capabilities in terms of flexibility of substrate materials. Section 4 and Section 5 describe the different flexible materials useful for antenna construction and the features notable in characterizing their selection, with polymer characteristics discussed in detail. Section 6 demonstrates the analysis and resultant comparison of the bending capabilities of different polymer-based flexible antennas on the radiation characteristics. Section 7 discusses the effects of bending on the resonant frequency, the reflection coefficient and the impact of loss tangent on the resonant frequency. The future outlook is presented in Section 8 and the findings of the paper are summarised in Section 9.

## 2. Advancement of Flexible Electronics

The history of materials used to provide flexibility in devices dates back more than half a century (see Table 1). Six decades ago, flexible single-crystalline silicon solar cells were implemented on satellites. These materials were bendable, non-breakable and shaped conformably [36,37]. A Thin Film Transistor (TFT) made of Tellurium was developed in 1968 on a piece of paper. In the same year, T.P Brody presented Mylar, Polyethylene and anodized Aluminum wrapping foils substrates [38]. In the mid-1980s, researchers achieved the highest ever curvature of any flexible electronic circuits.

During the 1980s, it was observed that the resulting circuit performance was not affected with as much as 1.6 mm in the curvature of the flexible substrate [94,95]. Polymers were deemed materials for insulating before the discovery of conductive polymers. After the discovery of Poly-acetylene in 1977, interest in conductive polymer materials in industries increased dramatically [99]. The development of Hydrogenated Amorphous Silicon Indium Tin Oxide (a-Si: H/ITO) cells on an organic polymer was another milestone in the advancement of flexible materials [100,101]. Japan developed Plasma Enhanced Chemical Vapor Deposition (PECVD) machines in the mid-1980s that provided a base for Si–H solar cell fabrication and led to the active-matrix liquid crystal display (AMLCD) industry in Japan. Moreover, in the 1990s, flexible polyimide was of interest because of its flexibility, low cost and thermal endurance. Constant et al. [53] fabricated a Si–H TFT circuit on a flexible polyimide substrate in 1995 which was significant because this was the first time photolithography was used to affix Si–H TFT on rigid silicon to form a polyimide film [103,119].

In 1996, a hydrogenated silicon Si: H/TFT was made on a flexible stainless-steel foil [103]. In 1997, a Polycrystalline silicon (Poly–Si) TFT was successfully applied to a plastic substrate using laser annealing technology. In 2005, a rollable electrophoretic display was presented by Philips and in 2006, Samsung announced a 7-inch flexible liquid crystal panel [108,109]. Furthermore, in 2006, Universal Display Corporation and the Palo Alto Research Centre presented a prototype flexible organic light-emitting diode (OLED) display which exhibited full colour and high resolution. It was constructed using a Poly–Si TFT backplane made on steel foil [110]. In 2008, Franky et al. [22] developed a White OLED (WOLED) on a 55-inch flat panel display using a vacuum deposition process. Then, in 2012, Indium Gallium Zinc Oxide (IGZO) was used as a backplane material which, advantageously, was compatible with flexible LCDs. Poor et al. [114] presented flexible displays on a plastic substrate in 2012 for smart Televisions.

Over the last seven years, there has been a boom in the world of flexible electronic and wearable devices. From 2013–2020, flexible electronic devices such as flexible smart sensors, RFID tags, flexible reconfigurable antennas, flexible energy harvesting circuits and wearable technology brought a remarkable change in healthcare, medical, industrial and entertainment sector. According to a report published by IDTech, forecasts from 2020–2030, indicated that the decade would be a great challenge for all companies producing wearable electronic devices including smartwatches, hearable aids, smart clothing, skin patches, Virtual Reality (VR) devices for general IoT applications [120]. Indeed, although IoT aims to connect everything to everything, the interface to this communication (the antenna) demands to be as flexible and easy to use as much as possible. A class of materials that allow flexibility and hence enable cheap and flexible deployment of antenna designs for faster and more practical IoT applications is further investigated here.

## 3. Flexible Antennas for General IoT Applications

IoT is the pervasive presence of various objects around us such as sensors, actuators, smartphones, smart computers, flexible antennas, RFID tags, and smartwatches, which can communicate with each other. Given the recent focus on IoTs and wearable flexible sensors, there is a new impetus for research into flexible electronics which can be bent or twisted so that they can be worn and mounted on various objects [121,122]. Flexible materials for various IoT applications require a high level of integrity of components and mechanical robustness with repeated rolling and bending capabilities. In particular, flexible smart fitness watches, RFID tags and wearable sensors are good examples of applications that use flexible material. Besides this, the elasticity and stretch-ability of materials are key properties required by electronic devices that require large and reversible deformation. These bendable devices also need to be versatile and may require an ability to store energy, operate with low power, and integrate with other devices and IoT applications [123,124,125], see Figure 3.

While most IoT projects have focused on long-range connectivity and low power usage, there has been some significant work, mainly in sensing, for short-range and extremely low power applications. Example short-range sensing on flexible material applications in IoT include:Applications [127,128] which require continuous location updates are an integral part of IoT as this provides the basis to monitor the object in the IoT system. Indeed, wearable trackers are used for tracking all manner of things such as human beings or animals [123,129].RFID technology [128,130], a unique identification system, is a prerequisite to deploying a smart device for sensing purpose.Energy harvesting technology [131,132] is a technology which is capturing usable energy from the environment to power the smart devices. The energy can be created by the variation of temperature, radio signals and the speed of the wind and stored in the form of capacitors.Sensors [133,134] are preliminary elements for IoTs. Variables which can be sensed include body temperature, blood pressure and heartbeat monitoring [124,135,136]. Flexible IoT sensors have been proposed as a way to monitor the healing of scars [137,138], body tumour detection [139,140], analyzing metabolites in the body [141], wearable flexible signalling system for astronauts [142] and flexible gait tracking sensors and non-contact vital sensors [143].An actuator [125,132] is also an important element of IoTs as it provides power to support movements in a system, such as controlling currents or the pressure of the liquid in the air. A piezoelectric actuator is a typical example of this which exhibits an electrical signal on the application of certain pressure onto it.

The interaction between objects through flexible IoT is illustrated in Figure 1, Figure 2, Figure 3, Figure 4. Although, IoTs aim to connect everything to everything, the interface to this communication, the wearable sensor, needs to be as flexible and as easy to use as possible. While the basic structure is the same for the flexible antenna and the chipless RFID tags, both structure and the radiation parameters are discussed in this section.

The Internet of Things (IoTs) is a growing number of physical objects that are connected wirelessly via antennas [144] which have implications for industrial automation, health and agriculture sectors, transportations and most importantly emergency responses to natural disasters. The IoT enables objects to share information and, when connected to decision-making algorithms, to make decisions accordingly. Nowadays, IoT devices are contributing to improving human lives and taking part in expanding the array of business applications. For instance, smart homes, in which residents can control the climate of rooms or turn on/off appliances such as TVs, alarm systems, monitoring systems and heating systems through remote access to their homes, automatically and wirelessly [134].

Virtually all current mobile electronic devices communicate incorporate passive antennas which are constructed with a variety of materials. There are many advantages of flexible antennas over traditional rigid antennas and the flexible antennas are gaining wider acceptance over the conventionally rigid materials because they can integrate lightweight, small thickness and low profile designs [16]. Compared to traditional antennas, there is a significant reduction in size and weight of flexible antennas by 50–70% and this size can be further reduced up to about 90% of its original size for a specific frequency when a flexible substrate, such as a polymer, replaces the hard printed boards [119]. Additionally, the flexible antennas are very robust yet lightweight and can withstand high mechanical strains [145,146,147].

The following sections provide a tutorial on the flexible antenna structure, bending capabilities and common flexible materials used for flexible electronics.

### 3.1. Basic Structure of Flexible Microstrip Patch Antenna

Flexible antennas consist of a conductive layer and dielectric material backing. As shown in Figure 4, the basic flexible microstrip patch antenna is a layer of thin conductive strip placed on the top of a flexible substrate. This conductive patch must maintain adequate conductivity even when it is stretched or deformed [145].

The properties of flexible antennas depend on their applications, required bendability and operating frequency. The different types of flexible materials which can be used as flexible substrates are analyzed and compared in this paper.

The essential structure of a flexible antenna is the same as a conventional antenna in that it has two major parts: a conductor and a dielectric. A conductive material is used as the radiating element or ground plane while a dielectric material acts as the substrate that supports the radiating element [142]. The basic parts of a flexible or wearable antenna are given in Figure 4. The general materials that can be used as a conductive layer include pure metals, metals mixed with fabrics and conductive inks where copper, aluminium, silver are examples of intrinsic metals and conductive polyester is a metal mixed fabric. Silver nanoparticles are an example of a conductive ink which allows an antenna to be printed on a substrate. Polymers, paper, foam, plastics, textile fabrics and soft PCB are popular dielectric materials. Figure 5 shows a breakdown of the different material types. The following subsections detail the conductive and substrate materials used for flexible antennas.

#### 3.1.1. Conductive Materials

Conductive materials are capable of transmitting electricity with little loss or radiating energy in the form of electromagnetic (EM) waves. These materials must be endowed with bending properties or, more specifically, should have the capability of bending, crumpling and stretching without affecting the antenna performance when used in applications that require the flexing of the antenna from its nominal straight configuration and should be resistant to material degradation. Moreover, the conductive material must be able to withstand repeated pressure while low resistivity and high conductivity maintain high importance in the electrical properties of these conductors. Table 2 depicts some important conductive material properties required for flexible electronic systems in general and as well as more specifically for flexible wearable devices such as flexible antennas and RFID tags where it is clear that conductive materials need a high level of conductivity, deformability and elasticity, as well as a high capacity for bending, adhesion and integration with textiles. They should simultaneously have a low level of resistivity, moisture and moisture absorption.

The materials used as the conductive or radiating part of an antenna can be categorized into three parts: (I) pure or intrinsic metals; (II) metal mixed fabrics; and (III) conductive inks. Intrinsic metals [3,10,148,149,150,151], aluminium [79] and silver paste [152] have all been used extensively in flexible antenna fabrication. These materials are highly conductive, cost-effective and provide little complexity in the fabrication process. In wearable electronics, a metal-plated textile is commonly known as an E-Textile. These conductive fibres, or E-Textiles, are widely used as conductive materials where ductile materials such as Kevlar, Nylon and Vectran are famous materials that can be coated with metals to form E-Fabrics with the benefit of being highly bendable and easy to sew [42,153,154]. The third category, the conductive inks, contain metallic particles such as carbon and silver nanoparticles which can be printed on a flex substrate with a standard printing process demonstrating that conductive inks are easy to fabricate and highly conductive. However, their limitations arise in high fabrication costs [7,14,26,81].

#### 3.1.2. Substrate Materials

The conductive layer of an antenna is fabricated on a substrate which determines the flexibility of the antenna. Where flexible substrates are more likely to bend and less tolerant of temperature variation, this equivalates to being less dimensionally stable as compared to non-flexible substrates and it is the dimensional stability of the substrate which affects the tenacity and fabrication of the conductive layer. While there are further challenges, including deformability, high thermal and electrical stability, moisture sustainability, lightweightedness and fabrication complexity which needs to be addressed while selecting a substrate for a flexible antenna, the selection of the substrate properties typically depends on its target application. Table 3 illustrates the general properties of substrate materials required by flexible devices.

In addressing high radiation performance in correspondence to a high level of flexibility, it is the mechanical, electrical and thermal properties of substrate materials which are very important. While in general, flexible substrates in flexible electronic systems and flexible antennas are characterized by their energy efficiency, lightweightedness, reduced fabrication complexities, mechanical robustness and low manufacturing costs, the cost of the antenna may increase in chasing radiation perfection. Although it is the flexibility of a substrate which dictates the bendability of an antenna, antenna deformation not only affects radiation patterns but also its resonant frequency, matching level, bandwidth, directivity, radiation energy and efficiency [7,81,149]. For flexible substrates to form an excellent alternative to current traditional rigid substrates, not only compactness but desirable radiation characteristics are also required for high radiation efficiency to be achieved [14,27,58].

## 4. Flexible Substrate Materials for Wearable Antennas

There are two major structural characteristics that form an antenna: a conductor and a dielectric, the essence of which is maintained for both the flexible and conventional antenna. The antenna operates by way of a conductive material used as a radiating element or a ground plane with a dielectric material to act as a substrate that supports the radiating element [142]. The materials that are commonly utilized for the conductive layer of the antennas include pure metals, like copper, aluminium and silver as examples of these intrinsic metals, metals mixed with fabrics like conductive polyester and conductive inks where silver nanoparticles are an example which allows an antenna to be printed on a substrate. Popular dielectric materials include polymers, paper, foam, plastic, textile fabrics and soft PCB.

Substrate materials are mainly used to support the conductive element of the antenna. Depending on the type of application, it is vital to choose a suitable flexible material as a substrate. In last few decades, various types of flexible materials have been adopted as substrates for antennas to provide flexibility including textile or fabrics [44,155,156,157,158,159], paper [160,161,162,163,164,165] and polymers [3,7,8,10,14,15,26,28,29,39,55,57,58,59,61,62,63,73,77,78,79,80,81,85,86,104,139,148,149,166,167,168,169,170,171,172,173,174,175,176,177,178,179,180,181,182,183,184,185,186,187,188]. Non-conductive textiles such as cotton, wool and silk, are combined with metallic conductive fibres to manufacture textile antennas [189,190]. The relative permittivity of these materials depends on the thickness and the nature of the fabric, whether knitted or woven [154]. Alternatively, the paper is a low cost and environmentally friendly material which can be modified to be fire retardant to make it suitable as a substrate for flexible antennas [165] while polymers are highly resistant nonconductive materials composed of repeated subunits of hydrocarbons known as monomers which are highly flexible, low cost, with low thickness and a low loss tangent [191]. Flexible polymer materials are used abundantly as substrates for flexible wearable devices.

Several papers providing extensive reviews on wearable antennas have been published [156,157,192,193,194,195,196,197]. In [192,193,194], antenna design, different materials, limitations concerning antennas and operation near the human body are reviewed. In [195], recent advancements in fabrication techniques for flexible antennas with a section on polymer substrate-based antennas are reviewed. In [156,157,196,197], the review is focused more specifically on textile and fabric-based antennas for wearable application.

The highly resistant nonconductive polymer materials, made up of repeated subunits of hydrocarbons known as monomers, are divided into two categories: Natural and Synthetic. Natural polymers include materials such as Silk, Rubber, Starch and Wool while the Synthetic polymers, which are prepared chemically in labs, include materials such as Polyvinylchloride (PVC), Polystyrene and Nylon. This section presents the different types of flexible polymers that have been investigated in the literature with a comparison of the factors that have an impact on the radiation parameters of microstrip patch antennas.

In this study, a complete survey on polymer-based flexible antennas application for wearable and general IoT applications is presented. To the best of our knowledge, such a comprehensive review does not exist in the published literature to date. Focused on flexible polymer-based antennas and the bending and moulding effects on the radiation characteristics of antennas, five different types of polymer material have been chosen that facilitate bending and flexibility as a substrate in an antenna of interest. These polymers are Polyimide (PI), Polyethylene Terephthalate (PET), Polydimethylsiloxane (PDMS), Polytetrafluoroethylene (PTFE), Rogers RT/Duroid and Liquid Crystal Polymer (LCP). The reason to choose these five materials is the fact that these polymers—with some concentration of other polymers—cover more than 80% of polymer-based FES manufacturing and the design of flexible antenna and RFID tags.

### 4.1. Polyimide (PI)

A Polyimide is a thin, flexible and lightweight polymer which is extensively used as core flexible material for supporting overlays for soft PCB processes such as PI Films [26,198]. PIs are widely used in flexible printed circuits (FPCs exhibiting their flexible endurance, excellent tear resistance, low dielectric constant, dissipation factor, moisture absorption and coefficient of linear thermal expansion (CTE). PIs provide numerous advantages over other flexible substrates where, for example, their thinness and flexibility make them useful for small area instalments and creating 3-D static shapes. The PIs are also utilised for making flat panel displays, chip packaging, antenna substrates, mobile phones, Smartwatches, video cameras, and notebook computers [199]. Polyimide is the most prescriptive material for IEEE 802.11 standards because of its high compatibility with signal processing circuits [169].

The Kapton Polyimide is a type of polyimide which is known for its flexibility and a good balance of physical, chemical and electrical properties (see Table 4). A reliable flexible substrate, Kapton is a low cost, has thermal endurance with mechanical robustness and a low loss factor over a wide range of frequencies [14]. Numerous compact Kapton-based flexible antennas are represented with different dimensions, conductive materials and range of frequencies [11,26,53,54,55,56] as presented in Table 3. In general, Kapton shows good soldering tolerance for flexible antenna fabrication as it withstands high temperature, which is a good feature for the thermal annealing of inkjet antenna printing [142]. The robustness and bending efficiency of Kapton PIs were subjected to a curvature test under the flexibility and bendability tests were performed and the effects of deformation (radii or curvature) on an antenna and its radiation characteristics evaluated [7,14,26,29,57,58].

Designs with Kapton-based flexible antennas were substantially generated. A compact Ultra-wideband (UWB) polyimide base flexible antenna is presented in [26] which, advantageously, shows very small susceptibility to deformation in terms of impedance matching and return loss. In [7], presenting a polyimide substrate antenna underwent convex and concave bending, no significant changes in the radiation pattern and frequency shift was observed. Coating with different materials has also been used to alter its physical or electrical characteristics where, for example, in [57], Kapton is coated with parylene–C, and no significant change in return loss and radiation energy when the antenna is subjected to bending is observed and similar results were observed in [29] where the Kapton substrate is doped with conductive polymer polyaniline (PANI) and undergoes vigorous crumpling. Two robust and flexible compact Kapton polyimide-based antennas are designed in [11]: the first antenna has a dual-band while the second is designed for a single band. Both antennas have good radiation characteristics and can be used for flexible displays with WLAN and Bluetooth connectivity. In [139], a UWB-16 array Kapton-based flexible antenna is pioneeringly proposed to detect breast cancer. The antenna was fabricated in a flexible bra and operated in the frequency range of 2–4 GHz which is within the bandwidth requirements used by Microwave Radar Imaging (MWI) for breast cancer. A wearable compact Kapton-based antenna is designed in [58]. The authors reported that after the antenna is deformed, it exhibits only minor changes in resonance frequency and return loss.

### 4.2. Polyethylene Terephthalate (PET)

Polyethylene Terephthalate (PET) film is a flexible, strong and somewhat rigid dimensionally stable thermoplastic polymer resin commonly known as Polyester. PET exists in both transparent (amorphous) and semi-crystalline form in accordance with its processing and thermal history [212]. An interesting material due to its thermal, electrical and moist stability and flexibility, PET has tremendous chemical resistive and physical properties [60,80,211].

PET film has been used extensively in applications, such as medical packaging, tape backing, printing films and flexible printed circuits, where substrate flexibility and transparency are required and is well known as the substrate for microstrip antennas, flexible chipped and chip-less RFID tags, fabrics and a wide range of textile and optoelectronics [30,61,62,63,65,168,181,182].

In [62], a chip-less RFID tag is designed with the Frequency Selective Surface (FSS) method and fabricated on PET and paper substrate. Dupont manufactured “Melinex 401” PET film was used in this design where an octagonal structure of silver paste was fabricated on it. Radar Cross Section (RCS) shows different combinations of bits 0s and 1s and were transmitted and received successfully from the assembly of two horn antennas. The tag is bent for different radii with results showing that there is a minor change in the RCS pattern, hence, successfully achieved encoded information. In [188], our new Bow-Tie design was designed, printed and compared to an octagonal-shaped tag that was published in the [62]. Both tags were designed using CST studio and fabricated on low-cost flexible PET substrate.

In [65], a Substrate Integrated Waveguide (SIW) flexible antenna is designed and fabricated on a PET substrate. The radiation characteristics of the antenna, measured for different bending levels, demonstrated, for the first time, PET substrate compatibility with SIW technology. The results show the applicability of SIW technology for low-cost wearable Internet of things.

A Millimeter-Wave (MMW) Flexible antenna on the PET substrate for 5 G applications is reported in [105]. The Coplanar waveguide (CPW) feed antenna, which is also applicable to non-polar surfaces and casual clothing, operates at Ka-band and promises a high gain with, for example, 8.2 dBi. In [66], a dual-band flexible reconfigurable antenna is fabricated on the PET polymer substrate where the proposed antenna can operate either on the single band at 2.36 GHz or dual-band with 3.64 GHz of the resonant frequency. The antenna demonstrates good radiation characteristics on being curved for both On/Off states of PIN-diode.

### 4.3. Polydimethylsiloxane (PDMS)

PDMS, commonly known as silicon, is an interesting polymer because of its softness and flexibility. PDMS has excellent rheological properties and is a flow-able, water-resistant, transparent and low-cost polymer with many attractive physical and chemical properties. Providing high deformability, flexible surface chemistry and low electrical conductivity [207,213,214], it is chemically inert due to its homogeneous and isotropic properties [215]. Indeed, PDMS has good chemical stability and a low dielectric constant between 2.3 to 2.8, see Table 4. The aforementioned advantages and excellent qualities have made PDMS a good substrate choice for stretchable and flexible antennas [68,69,70] for which it has, consequently, been used extensively as a substrate in electronic devices and microsystem fabrication [3].

PDMS properties can be changed by doping with other materials. More specifically, the mechanical, electrical properties and flexibility of a PDMS substrate can be changed by adding materials of different dielectric constants [148]. Additionally, PDMS, being stable at high temperatures which are required for processing of biological materials (40–95 °C), is attractive for applications requiring gradients [216,217]. Nevertheless, drawbacks associated with PDMS include fabrication complexity and a comparatively high cost against other flexible materials [3,39,71,72].

A commonly used PDMS is the Sylgard 184 Silicone Elastomer, a transparent polymer with good flame resistance, low Linear Coefficient of Thermal Expansion (CTE) and dissipation factor, high deformability and moisture stability but low electrical and thermal stability. PDMSs are extensively used in microfluidic systems, solar cells, industrial control systems, sensors, amplifiers, high voltage resistor packs and as a substrate in flexible electronics [207]. The general properties of Sylgard 184 are listed in Table 4.

PDMS-based flexible antenna designs have been published in numerous studies. The example flexible conformal antenna presented in [39] reports an embroidered conductive fibre material which is embedded over a ceramic PDMS substrate to get a high level of flexibility and stretchability. In [148], a Wireless Body Area Network (WBAN) and PDMS-based flexible antenna is shown to experience a very small frequency shift of resonant frequency after bending with a high level of robustness while, in [73], a flexible folded dipole slot antenna is presented over a PDMS substrate. The proposed dipole slot antenna design exhibits good flexibility over bending and minor effects on return losses.

### 4.4. Polytetrafluoroethylene (PTFE)

PTFE is a common polymer that has very interesting properties. PTFE is chemically stable, water-resistant and can withstand an astonishing degree of contrast in temperature with a low of −200 °C and high and a high of up to 250 °C [206]. Resultantly difficult to melt even at high temperature, it is very dense and while it is thermally stable, compared to other types of plastics, PTFE has low mechanical characteristics [205]. The main advantage of PTFE rests on its versatility, consequently being suitable for many applications where it is widely used in engineering and manufacturing and its moisture and oil repulsive nature, mean that PTFE is intensively utilised for storing corrosive materials [64,206,218]. The mechanical properties of PTFE can be modified by adding glass, graphite and carbon. For example, a doped PTFE might able to maintain its high temperature and chemical characteristics [205].

Teflon is a commercially available PTFE polymer having a combination of good mechanical, electrical, thermal and anti-friction properties. With a very high melting point where its functionalities continue over 260 °C [204], Teflon is a well-known flexible material because of this thermal stability and resistance to change in temperature and as well as its resistance to corrosion and stable dielectric constant over a wide range of frequencies [176]. Its properties serve to provide flexibility in antenna designs with PTFE Teflon aptly utilised in the RFID tag antenna designs. A brief comparison of PTFE properties with other materials is shown in Table 4.

Numerous research articles for PTFE substrate-based antenna designs have been published during the last few decades. In [77], a PTFE substrate-based slot antenna made of conductive textile is presented where, performing a bending analysis on the antenna over a spherical cylinder demonstrated that the antenna design exhibited high flexibility with a very small shift in the central frequency. Later, in [120], reflection characteristics are observed and measured in flat and spherically bent conditions by a flexible slot antenna over PTFE and Polyester substrates [78] with a good agreement of flat and flex design of the proposed antenna being reported with evidence that, compared to polyester substrate-based antennas, PTFE substrate-based antennas have improved radiation characteristics.

Rogers laminate, a type of PTFE, commercially available in the form of sheets, widely used these days, differ with typical PTFE material by dielectric constants but having similar mechanical properties. Thermally stable dielectric constants with an expansion coefficient that is quite similar to copper, some prominent qualities include easy fabrication on printed circuit boards, robustness, and excellent dimensional stability with typical etch shrinking [200,201,202]. Rogers Duroid substrates are another type of PTFE substrates that exhibits excellent matching and controlled impedance transmission over microwave circuits. This material is more rigid as compared to natural PTFE but very stable at high temperatures [200]. PTFE materials such as Rogers Laminates and Duroids are commercially available and have the lowest electrical loss for processed PTFE, uniform electrical properties over frequency, excellent chemical resistance and low moisture absorption [203].

In [79], the proposal of a flexible bow-tie antenna designed with Roger RO3003 substrate determined that the radiation patterns and return losses of the designed antenna were independent of the radius of curvature when the bending is larger than the antenna’s dimension. The effects of bending the Co-planar Waveguide (CPW) fed flexible bow-tie slot antenna are demonstrated in [15] to observe that, after attempting different bending levels of the antenna, there were only minor changes in resonant frequency and radiation patterns of the proposed antenna. This means that the bending has almost no effect on the proposed CPW-fed flexible bow-tie slot antenna design.

### 4.5. Liquid Crystal Polymers (LCP)

LCP, a special class of crystalline Aromatic Polyester based on monomers [219], consists of a series of thermoplastics that have a unique set of properties such as high heat resistance and tolerance, inherent flame redundancy and good weather sustainability. These LCPs have excellent chemical properties like their high anisotropy, which means flexibility, thermal expansion and stiffness are high in one direction. With good cycle repeatability due to their high melt flow [219,220], they also exhibit tremendous mechanical properties such as high strength, modulus of elasticity and toughness [220]. The mechanical strength and elastic modulus of LCP substrates are equal to or above that of other common plastics. They have a dense crystalline structure, an excellent electrical insulator and are resistant to arc at the flame of high temperature [221]. Table 4 shows a comparison of LCP general properties with other compatible flexible substrates used for antenna designs. Unfortunately, LCPs are expensive and their high anisotropy causes weakness at weld lines where the material meets different molecules [220]. Another disadvantage of the LCPs is that they are difficult to fabricate because their density denotes that they have small spaces as in little gaps in crystal composition such that processes like traditional etching which fabricate over crystal become difficult [81].

Despite the aforementioned LCP limitations, they are widely used as flexible substrates and have become one of the most desirable organic materials for high-frequency applications as they can withstand rises in operational frequencies and are very suitable for RFID antennas [82,83,84]. A design of series-fed two dipole antennas on the LCP substrate is presented in [81] where the bending effects on the proposed antennas are tested. The results demonstrated that bending has a minor effect on Voltage Standing Wave Ratio (VSWR) but evidenced decreased gain and directivity of the proposed antenna. In [8], a flexible dual-band LCP antenna is presented in [10] where radiation characteristics of the proposed antenna are measured at different bending angles and revealed that reflections are almost the same at bending angle of 60 degrees as compared to the flat condition of the antenna. A circularly polarized (CP) flexible CPW fed antenna is presented on the LCP substrate [85]. The antenna is tested at various bending angles (30°, 60°, 90°, 120°) at 3.5 and 5.8 GHz frequencies with reflection coefficients, at the different angles, showing a good correlation between simulated and measured values with just a slight shift in resonant frequency towards the lower component for bending angles at 60° and 90°. In [85], a CPW fed antenna is fabricated on an LCP substrate for WiMAX(3.5 GHz) and WLAN(5.8 GHz) applications and indicated good agreement of results for various degrees of curvature up to 120 degrees.

## 5. Comparative Analysis of Suitable Polymer Substrates for Flexible Antennas and RFID Tags

Table 5 depicts a comprehensive comparison between the antennas which are constructed from dielectric polymers PI, PET, PDMS, PTFE and LCPs. This section provides an overview of these polymers and the variants which are widely used as flexible substrates for antennas. All these materials have proven bending capabilities and the table serves to present a range of experimental verification with measured and simulated results that have been reported in different articles. Kapton is a variant of PI which is very common and widely used polyimide substrate for antennas which has high thermal and electrical stability. Sylgard 184 is commercially used as a flexible substrate which has high moisture stability and bending capabilities. Teflon, Rogers Laminates and RT/Duroid are common commercially available PTFEs that possess good thermal and electrical stability. Compared to all of the other materials reported, Rogers UTRALAM is classified as the most electrically and thermally stable LCP and has the additional features of being oil and water repellent.

Melinex 401 CW is a slippery surfaced, transparent and highly dense film which has many outstanding features over the other available substrate films: it has a high dielectric constant and tensile strength and a low dissipation factor and its temperature Co-efficient of Resistance (CTE), therefore it exhibits high thermal and electrical stability and low shrinking capabilities, as shown in Table 4. It does, however, have low deformability and moisture absorption capability, compared to other substrates, making it less efficient for highly flexible applications. It can be observed that Sylgard 184 is an antenna substrate with low density (≤1 g/cc) which possesses high deformability at standard temperatures and that the tensile strength of Kapton, Rogers Laminates and LCP are high (e.g., ≥5 Kpsi) and, therefore, comparatively more thermally and electrically stable.

Teflon and Sylgard 184 are classified with low dielectric constant (≤3 at 100 Hz–1 MHz) and low dielectric strength (≤1000 v/mil) while possessing high radiation characteristics and, as conveyed in Table 4, high deformability can indeed be achieved for low density and tensile strength. The low Coefficient of Linear Thermal Expansion (CTE ≤ 30 ppm/C) of substrate materials such as PI, PTFE and LCP serve to make them more thermally stable and, furthermore, it is observed from Table 4 that all the substrates which have very low dissipation factors or loss tangents (tanσ ≤ 0.01 at 100 Hz–1 MHz) lead towards electrical stability. The low values of moisture absorption, usually ≤0.1%, define the moisture stability of materials whilst, although PTFE and LCPs are highly moisture resistant, they are contrastingly difficult to fabricate as compared to other material.

## 6. Deformational or Bending Effects of Flexible Polymer Substrates on Antenna Radiation Characteristics

Table 5 is a major collection of the most significant and recent experiments on bending characteristics of these five polymer materials and antennas. A total of 50 experimental works are analysed on flexible antennas and we selected 20 experimental works for qualitative analysis to give the reader a comprehensive overview of the entire area for the range of different frequencies and applications. These experiments are carried out in a different environment and with different fabrication and experimental procedures. However, knowing the limitations, we grouped some experiments for the same frequency ranges and almost the same physical properties, see Section 6. In addition, the reflection coefficients, gain and the shifts in the resonant frequencies at various bending states for flexible polymer substrate-based antennas from the literature are compared (see Table 5). In this table, polymers such as PI, PET, PTFE, PDMS and LCP with some varients are analysed from the previous articles. It also provides a comparative analysis of the radiation characteristics for a specific application and indicates the impact of bending on return loss, resonant frequency and the gain of the antenna.

The most extensively 21tilized polymer amongst these five polymers is PI as a substrate of flexible antennas. Kapton is a commercially available variant of PI. The choice of Polyimide Kapton as a substrate of the antenna was due to its good physical balance, chemical, electrical properties and high thermal stability. Kapton (200 HN) with 50.8 μm thickness is used as a substrate having dimensions of 45 × 30 mm, 47 × 33 mm and 30 × 33 mm with silver nanoparticles as the conductive material in [14,26,58]. In [14], it was observed through bending analysis at the curvature of 13 mm, that the shift in resonant frequency is 80 MHz, which is 0.36% of the resonant frequency of 2.45 GHz, with a return loss of ± 1 dB. In [26], the highest 0.4% shift in the resonant frequency compared to the flat case is observed towards the first resonance frequency which is less than 50 MHz. In [58], Kapton (200 HN) is used as a substrate with silver nanoparticles as a conductor, the highest frequency shift of 1.1 GHz is obtained when the antenna is curved to 9 mm with a smaller dimension and return loss increase by −3 dB due to an increase in directivity at 8.8 GHz. Similarly, Kapton (500 HN), with an increased thickness of about 130 µm, is used in various flexible antennas as it has a higher dissipation factor and less resistivity compared to Kapton (200 HN) [209]. A compact Kapton-based inkjet-Printed Multiband flexible antenna is reported in [7] on which convex and concave bending performed at a maximum level of 59 mm. The antenna covers four-wide frequency bands centred at 1.2, 2.0, 2.6 and 3.4 GHz. In convex bending, no significant shift of frequency is observed but in concave bending, a 3% frequency shift is detected towards the lower end with all frequency bands. At higher central frequencies, the gain is increased with convex bending of up to 59 mm due to the increase in directivity when the antenna undergoes this higher curvature. A compact flexible antenna for WLAN and upper UWB applications is presented in [57] in which Kapton 500 HN is coated with Parylene–C to increase flexibility without much effect on return losses and gain. Wearable flexible antennas may undergo crumpling, which can affect their radiation performance. In this study, Kapton 500 HN is shown to be an efficient polymer which withstands a high crumpling level of up to 5.5 mm. A Dual-Band Elliptical Polymer antenna that uses a flexible Kapton substrate doped with conductive poly is presented by Hamouda et al [29] who reported that the crumpling has an effect on the proposed antenna’s performance at a high operating frequency of 5.8 GHz with a maximum gain of 2.48 dBi.

PET film has extensively been used in applications where flexibility and transparency of substrate are required. In [66], a reconfigurable folded slot antennas antenna is fabricated on PET substrate and its radiation characteristics are observed at ON and OFF state. At ON state, the antenna is a single band at 2.42 GHz and with the curvature of 25 mm, its resonant frequency only shifted by 0.1%. Similarly, in [62], an RFID tag is fabricated on PET substrate and its RCS is observed over various bending states. It is observed that a working bent tag up to 16 mm a read range of 3.5 m was successfully achieved. These results are graphically analyzed in the discussion part of this paper.

PDMS is a well-known polymer because of its tremendous rheological properties as it is a flow-able, water-resistant, transparent and low-cost polymer with many attractive physical and chemical properties. The PDMS polymer Sylgard 184, because of its excellent bending features, is used as a flexible substrate in various proposed designs. It is used as a substrate with 2 mm thickness and dimensions of 50 × 40 mm (fibre tissue) in [170] and 130 × 80 mm (patch) in [148]. As a fibre tissue, the proposed Transparent Flexible Polymer Fabric Tissue Antenna in [170], undergoes bending levels maximum at 25 mm with measured results showing that the resonant frequencies are shifted towards higher components about 0.085% of operating frequency at 7 GHz and 0.25% of operating frequency at 17.5 GHz. While the return losses fluctuate because of the change in directivity when the antenna is in bending state, for flexible W-BAN antennas, the frequency shift is negligible for all curvatures up to 230 mm [148] which could be due to the small bending curvature.

PTFE is a polymer which is mostly used in flexible electronics such as Teflon and Rogers laminates. However, it has limited use in flexible antenna designs because PTFE has relatively low deformability and even less electrical stability to bend (see Table 4). In [78], a comparative analysis of PTFE substrate antennas with polyester fabric substrate antennas is subjected to curvature. A significant effect on the antenna performance is observed at the bending level of 225 mm on the PTFE substrate antenna and causes a minor shift in resonant frequency. However, the same PTFE substrate, with the same dimension of 95 × 90 mm but a thickness of 500 µm, instead of 127 µm, the central frequency is shifted by 84 MHz towards a lower frequency at 200 mm radial curvature and leads to a reduction in the bandwidth from 239 MHz to 162 MHz and an increase of the reflection coefficient (S_11_) to −10 dB [77].

LCP is a highly anisotropic special class of crystalline polymer which is based on monomers. It has good chemical properties which are suitable for various flexible antenna applications such as in [10], a dual-band antenna is fabricated on LCP substrate. The fully flexible antennas were developed to operate at 2.45 and 5.8 GHz frequencies, showing identical reflections for the bend up to 60 degrees or 3.5 cm of a radius of curvature.

## 7. Discussion on the Existing Work

To investigate the bending behaviour of flexible antennas, some challenges and limitations need to be considered, such as substrate thickness, dimensions of antenna, feeding techniques, physical properties of materials and mostly the frequency range for which antenna is designed. However, we categorised flexible antennas based on five polymer materials PI, PET, PDMS, PTFE and LCP from the recent and noticeable literature and grouped them together for almost same physical properties and comparable dimensions. Furthermore, we divided them into three range of frequencies 2.2 to 2.5, 2.5 to 5.0 and greater than 5.0 GHz. The reason to investigate antennas for specific ranges of frequency is to avoid abrupt changes in radiation characteristics of antennas which could make analysis unrealistic. Despite this, there is an impact for change in frequency for different antennas for the same substrate material, as it is not adequate for a specific range frequency and hence we could investigate them to get few results and recommendations for future endeavour.

Table 5 shows the effect on performance that the bending of antennas has when using different substrate materials. The comparison of the bending effects on resonant Frequency Shifts (FS) is registered as a percentage (%) and signal strength in terms of reflection coefficient (S_11_) have been analyzed and discussed in this section. In order to understand the behaviours of flexible polymers in terms of frequency shifts, the operating frequency bands are divided into three ranges for different applications, e.g., 2.2–2.5 GHz, 2.5–5.0 GHz and 5.0–30 GHz with bending ranging from 200 to 6 mm; see Figure 6a–c.

### 7.1. Effect of Bending on Resonant Frequency

Comparing the entries in Table 5, it can be deduced that the effects of bending or curvature of different polymer substrates have a different impact on FS at different levels of bending [80]. The resonant frequency and the return loss are analyzed for polymer-based antennas for curvature up to 6mm. Figure 6a–c present the average shifts from the central Frequency (F) ranging from 2.2 GHz up to 11 GHz.

Figure 6a shows the average shifts from the central frequencies for the band 2.2–2.5 GHz, which is the commonly used frequency band for Industrial, Scientific and Medical (ISM) applications, such as Wi-Fi and Bluetooth. In this context, PI is less affected, with an average of only 2.93% of this central frequency with the highest Shift (HS) of 6.52% and Lowest Shift (LS) of 0.45%, by bending from 200 mm to 6 mm curvature Radius (R). In contrast, the PTFE-based antennas are highly impacted by bending with an average frequency shift of about 11.7% (HS: 18.88%, LS: 14.14%).

Alternatively, as illustrated in Figure 6b, for the operating frequency range of 2.5–5.0 GHz which lies in X-Band and is suitable for WiMAX applications, PTFE-based flexible antennas prove to be more efficient by providing an average frequency shift of approximately 4% (HS: 10.25%, LS: 0.56%). On the other hand, as depicted in Figure 6c, PTFE and LCPs are very efficient in this higher range of frequencies, from 5.0–30 GHz, where less shift in frequency is observed with about 0.138% and 1.57% on average respectively.

### 7.2. Effect of Bending on Reflection Co-Efficient (S11)

One of the impacts of bending on an antenna’s performance is the reflection or scattering of waves and resultant impedance mismatch. While an impedance miss-match should occur when a flexible antenna is bent and crumpled, this simultaneously leads to a change in signal strength which often decreases [223], although sometimes increases, after bending the antenna. Figure 7a–c represents the twisting effects of an antenna on S_11_.

In order to analyse the effects of bending on impedance mismatch or reflection coefficient (S_11_), the average of S_11_ is evaluated from a flat case to a bend level up to 6mm for different ranges of frequencies. Figure 7a–c represent the percentage deviation of reflection co-efficient for flexible antennas with polymer substrates. PI and PTFE are the least effected flexible antenna polymer substrates. They demonstrate an average of less than 2% change in S_11_ for frequency ranging from 2.2 to 2.5 GHz, see Figure 7a. In variance, for frequencies ranging from 2.5 to 5.0 GHz, while PTFE-based antennas have better signal strength after bending, the PET-based antennas have a high average signal degradation and mismatch of about 13.15%, see Figure 7b. Meanwhile, for the range of frequencies higher than 5 GHz, both the PI- and LCP-based antennas experience lesser impact on S_11_, about 2% and 2.8% respectively, and an increased average signal strength, see Figure 7C. In this context, PTFE is badly impacted by approximately by 23.3% change in S_11_, which shows a lot of signal degradation when it is bent up to 6mm on average for frequencies higher than 5 GHz, see Figure 7C.

## 8. Future Outlook

The review covered in this paper is mainly related to flexible, wearable antennas based on the polymer materials used to provide flexibility for various FES. The area of investigation includes the primary and secondary research, focusing on the most efficient and relevant polymer substrates in terms of bending and flexibility, which is a key challenge for flexible IoT and wearable applications and includes polymers such as PI, PET, PDMS, PTFE and LCP which cover more than 90% of the polymer-based flexible wearable industry. This review gives emphasis on a detailed comparative analysis of the physical, electrical, thermal and chemical properties of the flexible polymer materials which have been used as substrates during the last few decades to provide flexibility, which, in the process of appropriate substrate selection for flexible antennas, simultaneously serves as a beneficial guide to better match their compatibility with specific wearable applications. It also provided an analysis of the bending effects on radiation characteristics of the flexible polymers PET, PTFE Teflon and PVC as substrates for antennas operating in three different frequency ranges: (I) of 2.2–2.5 GHz, (II) 2.5–5 GHz and (III) greater than 5 GHz.

In this comprehensive review, the bending capabilities of various polymers used as substrates for wearable antennas were compared for the different range of frequencies. In this study, the flexibility analysis is categorised for three frequency groups, the first group 2.2–2.5 GHz is important for ISM applications, second group 2.5–5 GHz is perfect for Radars, Mobile Phones, and Commercial Wireless LAN applications and the third group, greater than 5 GHz is essential for 5 G mobile communication. Understanding the fact that just a small change in operating frequency might impact the radiation characteristics in general and S-parameters and impedance mismatch in specific, the three groups of frequencies were selected. The first group 2.2–2.5 GHz, was small enough to see the impact of bending, however, the size of second (2.5–5.0 GHz) and the third group (greater than 5 GHz) of frequency ranges could be reduced to obtain a more specific analysis of bending capabilities. Hence in future, the small size of frequency ranges or increase in the number of groups with small bandwidths is a recommendation to investigate polymer substrate flexibility more accurately.

Based on the analysis of the polymer substrate antennas presented in Section 4, PI, PET, PDMS, PTFE and LCP are the five most abundant flexible polymer substrates selected for bending analysis and practical implementation of PET, PTFE Teflon and PVC substrate were selected for flexibility testing. In future, based on the comprehensive comparative analysis of the properties of polymers substrates presented in this paper, some other polymers or some variants of these five polymers with comparable or mostly suitable physical, electrical, mechanical and thermal properties can be used to advance the testing of materials for the bending capabilities of wearable antennas and RFID tags.

As the bending of flexible antennas highly impacts its S-parameters and impedance matching, which are crucial for receiving and transmitting the information, this first part of the study gives emphasis to these considerations by more specifically regarding the bending impacts on S-parameters impedance mismatch, and the signal degradation. The bending of flexible antennas, however, also has an impact on other parameters such as directivity, gain, radiation pattern and polarization and in future, the span of this study could be broadened to further consider the effects on these other parameters.

## 9. Conclusions

This comprehensive review gives emphasis on a detailed comparative analysis of the physical, electrical, thermal and chemical properties of the flexible polymer materials which have been used as substrates during the last few decades to provide flexibility, which, in the process of appropriate substrate selection for flexible antennas, simultaneously serves as a beneficial guide to better match their compatibility with specific wearable applications. The flexibility of IoTs is an important aspect which in turn requires flexible antennas for communication. Polymers such as Polyimides, Polyethylene-Terephthalate, Polydimethylsiloxane, Polytetrafluoroethylene and Liquid Crystal Polymers are widely used in flexible electronics and flexible antenna designs and cover more than 90% of the polymer-based flexible wearable industry. In this paper, we also present a comparative analysis of the properties of various polymer materials and an overview of the bending capabilities of polymer-based flexible antennas which can be used for flexible IoTs. Radiation characteristics of flexible antenna such as return losses, gain and directivity are drastically affected when the proposed antenna undergoes degradation or curvature. To corroborate the impact of bending of a flexible microstrip patch antenna, the flexible antennas for operating frequency ranges from 2–2.5 GHz, 2.5–5 GHz and greater than 5 GHz, on polymer substrates such as PI, PET, PDMS, TEFLON and PVC are critically observed to understand the bending capabilities. Herein, the behaviour of these flexible antennas with different substrates, for several folding conditions with a curvature radius down to the lowest value of 7 mm were analysed. It is concluded that the radiation characteristics have a huge impact when the antenna undergoes bending. Beyond a certain level of curvature, radiation is too distorted to obtain good results.

## Figures and Tables

**Figure 1 polymers-13-00357-f001:**
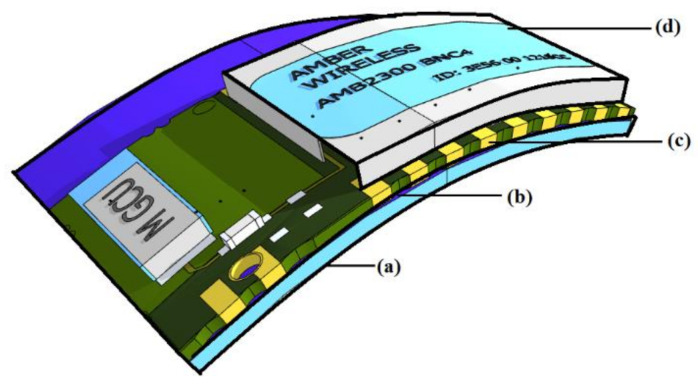
Representation of a Flexible Electronic System (FES) with four major components (**a**) Flexible Substrate (**b**) backplane (**c**) Front Panel (**d**) Encapsulation.

**Figure 2 polymers-13-00357-f002:**
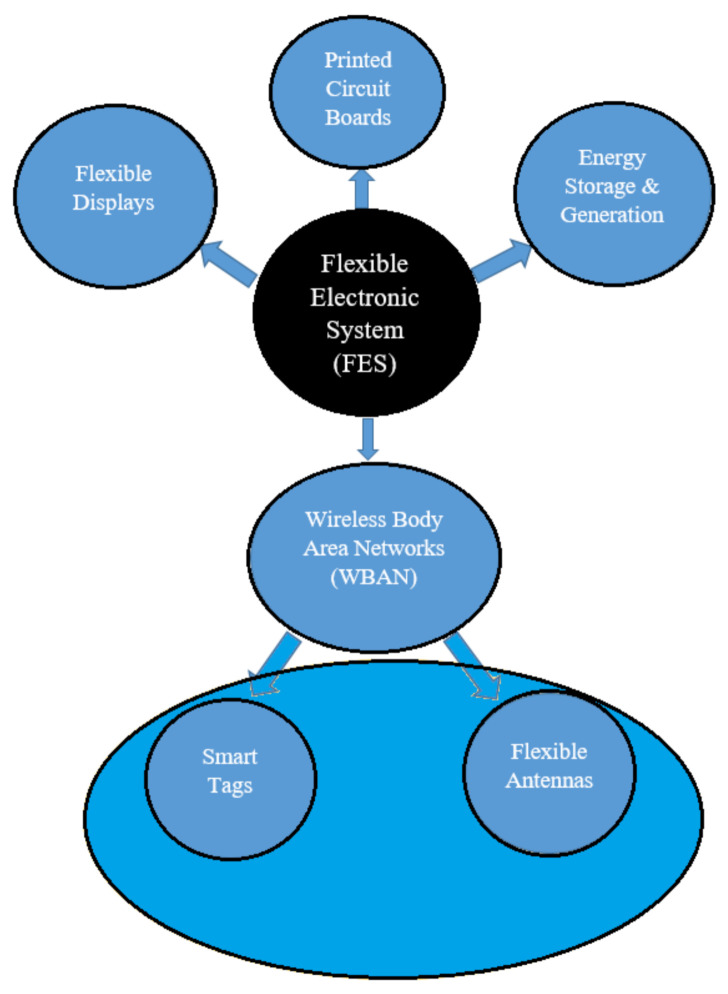
The next generation flexible electronic system (FES).

**Figure 3 polymers-13-00357-f003:**
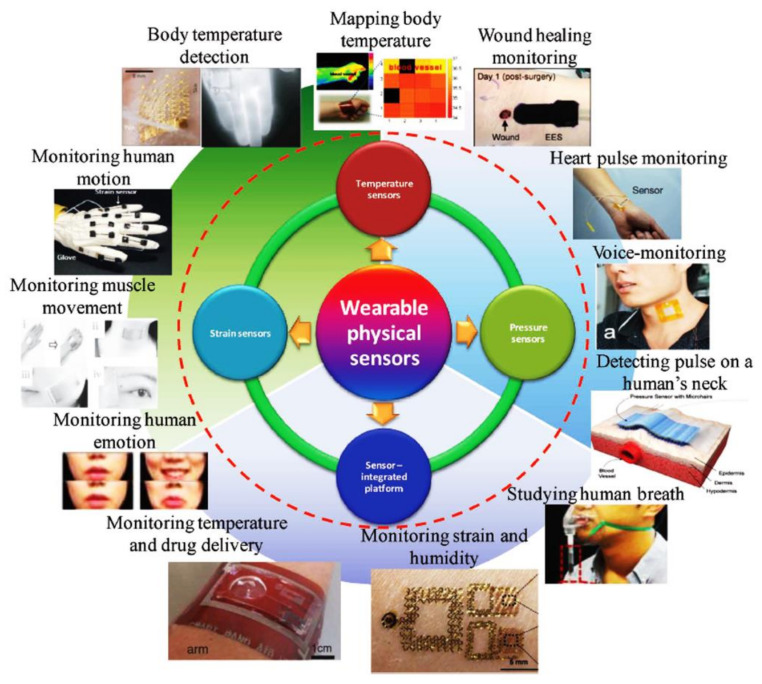
An overview of Flexible Wearable Internet of Things (IoTs) [126]. Courtesy of John Wiley and Sons, Ltd., Hoboken, NJ, USA.

**Figure 4 polymers-13-00357-f004:**
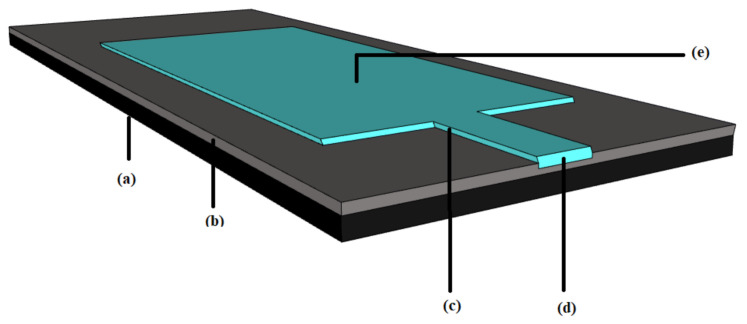
The general structure of flexible microstrip patch antenna (**a**) Ground plane (**b**) Flexible substrate (**c**) Transmission line (**d**) Feed (**e**) Conductive patch.

**Figure 5 polymers-13-00357-f005:**
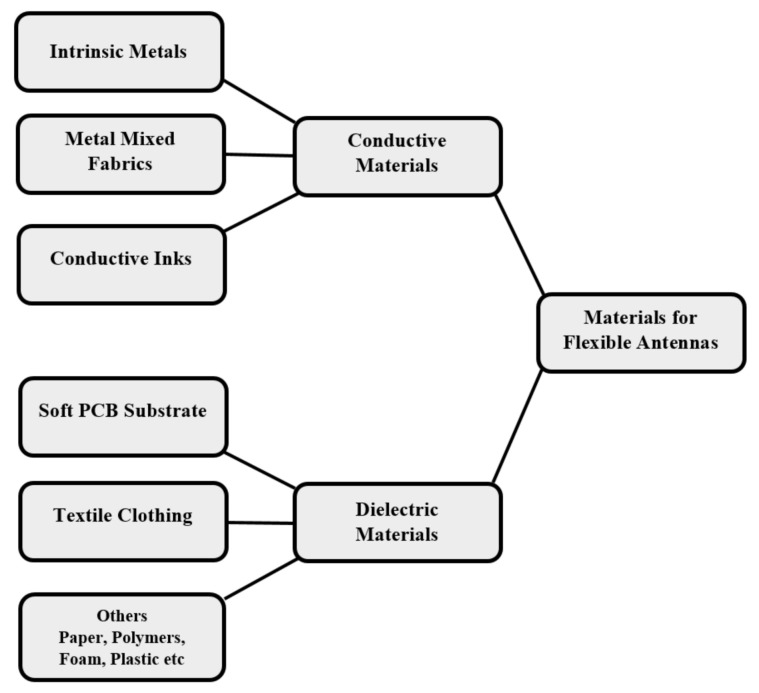
Materials for flexible antennas.

**Figure 6 polymers-13-00357-f006:**
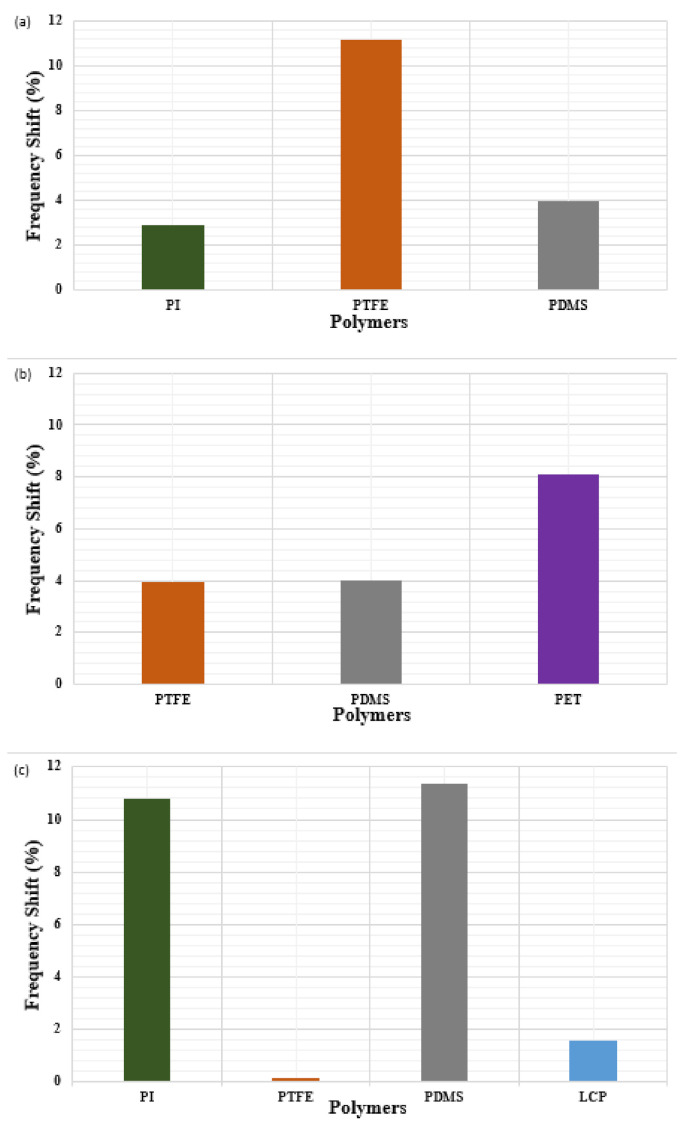
Average Frequency Shift in central frequency for (**a**) Polyimide (PI), Polydimethylsiloxane (PDMS) and Polytetrafluoroethylene (PTFE) that operate at a frequency range of 2.2–2.5 GHz, (**b**) PTFE, PDMS and Polyethylene Terephthalate (PET) that operate at a frequency range of 2.5–5.0 GHz, (**c**) PI, PTFE, PDMS and Liquid Crystal Polymer (LCP) that operate at frequency range greater than 5 GHz.

**Figure 7 polymers-13-00357-f007:**
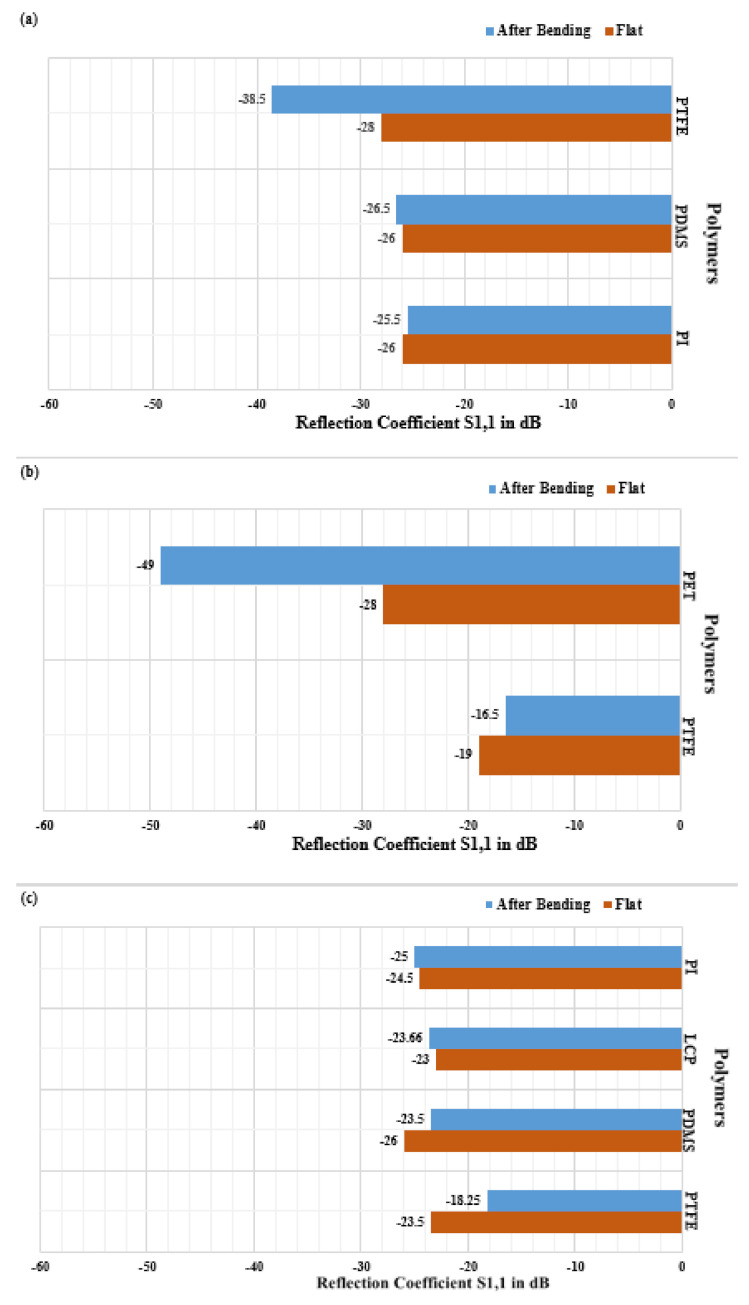
Average change in S11 in dB of polymer substrate-based antennas for (**a**) PI, PDMS and PTFE that operate at a frequency range of 2.2–2.5 GHz, (**b**) PTFE, PDMS and PET that operate at a frequency range of 2.5–5.0 GHz, (**c**) PI, PTFE, PDMS and LCP that operate at frequency range greater than 5 GHz.

**Table 1 polymers-13-00357-t001:** Chronological advancement in flexible electronics.

Advancement in Flexible Materials	Year/Era	References
Early use of flexible material as straw with slurry for strengthening houses	BC	[89]
Discovery of electric conduction in organic materials	1862	[90]
Polymers conceptualized—rapid development followed	1920	[91]
Invention of flexible solar panels	1967	[92]
Implementation of flexible single silicon solar cells on satellites	1968	[92,93]
Development of Thin Film Transistor (TFT) by Radio Corporation America (RCA)	1968	[94,95]
Development of the first Liquid Crystal Display (LCD)	1973	[96,97]
Implementation of Hydrogenated Amorphous Silicon (a-Si:H) cells on flexible polymers	1976	[98]
Development of the conjugated polymer “Polyacetylene”	1977	[99]
Development of a-Si:H/ITO cells on organic polymer	Early 1980’s	[100,101]
Invention of Active Matrix Liquid Crystal Displays (AMLCD) in Japan	Mid 1980’s	[100]
Organic Light-Emitting Diodes (OLED) display on the flexible substrate	1992	[102]
Development of a-Si:H/TFT circuit on flexible polyimide	1994	[53]
Integration of OLED with a-Si TFT on a metal foil	1996	[103]
Development of a-Si:H TFT on the flexible stainless steel foil	1996	[103]
Development of Polycrystalline Silicon (Poly-Si) TFT on a plastic substrate	1997	[104,105]
Implementation of a multilayer inorganic and polymer substrates	2003	[106,107]
Phillips produced rollable electrophoretic displays	2005	[108]
Samsung developed a 7” flexible LCD	2006	[109]
Universal Display Centre and Palo Alto Research Centre presented OLED displays with full colour	2006	[110]
Development of polymer hybrid material for permeation barriers	2008	[111]
Development of White Organic Light Emitting Diode (WOLED) displays	2008	[112]
The development of the first flexible smartphone called paper phone	2011	[113]
Development of flexible displays on plastic for smartphones	2012	[114]
Curved OLED Display for 55-inch television and smartphones	2013	[115]
Flexible paper display for eBooks	2013	[116]
Flexible erasable writeable paperless tablet using LCPs	2013	[116]
AMOLED flexible display technology	2013	[117]
Flexible smartwatches, flexible heartbeat and blood pressure measuring sensors, RFID tags, flexible reconfigurable antennas, flexible energy harvesting circuits,	2013–2020	[118]

**Table 2 polymers-13-00357-t002:** General requirements of conductive materials for flexible devices.

Properties	Level
Resistivity	Low
Conductivity	High
Deformability	High
Bending/Crumpling	High
Adhesion	High
Moisture absorption	Low
Elasticity	High
Environmental Degradation	Low
Integration with textiles	High
Fabrication complexity	Low

**Table 3 polymers-13-00357-t003:** General requirements of substrate materials for flexible devices [142].

**Mechanical Properties**	**Elastic Modulus**		**Stiffness**		**Deformability**	
	**High**		**Low**		**High**	
**Electrical Properties**	**Electrical Insulation**		**Dissipation Factor**		**Electrical Stability**	
	High		High		High	
**Thermal Properties**	**Coefficient of Linear Thermal Expansion (CTE)**		**Moisture Absorption**		**Thermal Stability**	
	Low		Low		High	
**Other Properties**	**Chemical Inertness**	**Weight**	**Fabrication Complexity**	**Surface Roughness**	**Opaqueness**	**Cost**
	High	Low	Low	Depends on application	Depends on application	Low

**Table 4 polymers-13-00357-t004:** Comparison of general requirements of different flexible substrate for antennas [200,201,202,203,204,205,206,207,208,209,210,211].

Substrate Model/Version	Physical/Mechanical Properties			Electrical Properties			Thermal/Chemical Properties			Comparative Analysis				
	Density (g/cc)	Tensile Strength X-Direction at 23 °C (Kpsi)	Tensile Modulus X-Direction at 23 °C (Kpsi)	Dielectric Constants 100 Hz–1 MHz	Dielectric Strength (V/mil)	Tanσ at 100 Hz–1 MHz	CLTE −15 C to 300 C (ppm/°C)	Moist Absorption (%) at 23 °C	Shrinkage (%) 30 min, 150 °C	Deformability	Thermal Stability	Moist Sustainability	Electrical Stability	Fabrication Complexity
**PI**	1.42–1.53 High	22–33 High	330–400 High	3.4–3.9 High	3500–7000 High	0.0013–0.0040 Low	20 Low	1.3–2.5 High	0.03–1.25 Low	Low	High	Low	High	Simple
Kapton														
HN,FN,HPP-ST														
PET	1.3–1.4 High	25–40 High	280–580 High	3.0 High	4000–4500 High	0.002 Low	19–20 Low	0.1–0.7 Low	0.5–1.1 Low	Low	High	High	High	Moderate
Melinex 401														
Polyester														
PDMS	0.97 Low	0.25–1.3 Low	0.522–0.126 Low	2.3–2.8 Low	342–551 Low	0.0015–0.0035 Low	340 High	0.1 Low	0.03 to 2.7 High	High	Low	High	Low	Moderate
Sylgard 184														
PTFE	2.1–2.25 High	3.9–4.1 Low	50–90 Low	2.1–2.72 Low	285 Low	0.0002–0.0025 Low	250–275 High	0–0.05 Low	1.5–3.0 High	Low	Low	High	Low	Complex
Teflon														
PTFE	2.0–3.0 High	20.3–29.5 High	65–300 High	3.0–10.2 High	780 Low	0.0004–0.0035 Low	10–17 Low	0.02–0.05 Low	0.05–0.1 Low	Low	High	High	Low	Simple/Printable
Rogers Laminate														
RO3000														
RO4000														
RT/Duroid														
LCP	1.4 High	29.0 High	327 High	2.9–3.14 High	3500 High	0.0025 Low	17 Low	0.04 Low	0.03 Low	Low	High	High	High	Complex/Non printable
Rogers ULTRALAM L														
3000														
LCP														

**Table 5 polymers-13-00357-t005:** Comparison of the effects of curvature on the radiation characteristics of different flexible substrates for mictrostrip patch antennas.

Substrate	Versions	Conductive Material	Ground Plane	ϵ_r_	Loss Tangent tanσ	Substrate Dimension (mm)	Substrate Thickness (μm)	Bending Level Curvature (mm)		Band/Frequency F_o_ (GHz)		Return Loss S_1,1_ (dB)		Gain (dBi)			Application
**Polyimides (PI)** [7,14,26,29,57,58]	Kapton 200 HN (2 mil 50 μm)	Silver Nano Particles	AMC	3.5	0.002	45 × 30	50.8	Flat		2.45		−29		NA			ISM
								r_1_ = 27 mm		2.35		−30					
								r_2_ = 13 mm		2.30		−28					
	Kapton 500 HN (5 mil 125 μm)	Silver Nano Particles	CPW Curved shape	3.5	0.0026	70 × 70	110	Convex	Concave	Multiband		Gain dBi Convex		Gain dBi Concave			WLAN
								r_1_ = 78 mm	r_1_ = 78 mm	Bands		r_1_	r_2_	r_1_	r_2_		ISM
								r_2_ = 59 mm	r_2_ = 59 mm	(1) 0.87–1.0		−1.2	−1.2	−1.1	−1.1		Bluetooth
								Unbent		(2) 1.4–2.2		0.9	1.0	0.9	0.9		LTE
										(3) 2.5–2.7		2.3	2.5	2.1	2.2		WiMAX
										Unbent		(1) −1.2	(2) 0.6		(3) 2.1		
	Kapton 200 HN (2 mil 50 μm)	Silver Nano Particles	NA	3.4	0.002	47 × 33	50.8	Flat		ISM/UWB		S_1,1_(dB)					ISM UWB
										2.2–14.3		Flat	r_1_		r_2_		
								r_1_ = 10 mm		2.5–5.5		2212−23	−24		−22		
										4.9–8.6		−37	−29		−27		
								r_2_ = 8 mm		10–14.3		−23	−24		−27		
	Kapton 500 HN + parylene–C	Copper	Copper	3.5	0.0008	32 × 18	127	Flat		9 GHz		−35 dB		NA			WLAN
				2.95			10	r = 40 mm				−32.5 dB					UWB
	Kapton 500 HN	PANI + MWCNT	NA	3.48	0.0026	48 × 33	130	Unrumpled		4.2		−17		NA			PCS
						35 × 33 (crump)		Crumpled at r = 5.5 mm		1.9, 5.7		−18,−26					WLAN
	Kapton 200 HN (2 mil 50 μm)	Silver Nano Particles	NA	3.4	0.0020	30 × 33	50.8	Flat		8.8		−23		NA			UWB
								r_1_ = 9 mm		7.7		−26					
								r_2_ = 7 mm		8.2		−24					
**Polyethylene Terephthalate (PET)** [62,66]	PET Polymer	Copper	Copper	3.0	0.008	59 × 4.75	100	Flat (Switch: OFF)		2.35, 3.61		−15, −12.5		NA			Dual Band
								r = 25 mm (Switch: OFF)		2.34, 3.64		−18, −15					
								Flat (Switch: ON)		2.41		−16					Single Band
								r = 25 mm (Switch: ON)		2.44		−21					
	Melinex CW 401	Micro-silver particles	NA	3.0	0.002	52 × 82	50	Flat		4.2, 5.3, 6.4, 7.5, 8.8		−28		NA			RFID RCS UWB
								r = 100 mm		4.4, 5.4, 6.5, 7.6, 8.9		−47					
								r = 60 mm		4.4, 5.3 6.5, 7.4, 8.8		−49					
								r = 30 mm		4.5, 5.6, 6.7, 7.4, 9.6		−50					
								r = 16 mm		3.6, 4.3, 6.5, 7.6, 9.0		−50					
**Polydimethylsiloxane (PDMS)** [3,73,148,170]	PDMS Sylgard184	Copper	Copper	2.65	0.02	46.4 × 20	2000	Flat		6.35		−36 dB		NA			Wire-Less
	SU-8			2.9	0.04			Curvature K = 66.91		6.0		−17 dB					
	PDMS Sylgard184	Fibre Tissue	Fibre Tissue	2.85	0.02	50 × 40	2000	Flat		7.0	17.5	−26	−30	4.3		4.5	UWB
								r_1_ = 50 mm		8.0	21.0	−19	−17	3.9		3.7	
								r_2_ = 25 mm		7.8	18.2	−28	−23	3.2		3.3	
	PDMS Sylgard184	Copper	NA	2.65	0.02	130 × 80	2000	Flat		381		−15		NA			TETRAPOL (385 MHz)
								r = 230 mm		383		−18					
	PDMS	Nickel	NA	2.2	0.013	-	5000	Flat		2.22		−28		NA			ISM Band
		Gold						r_1_ = 80 mm		2.24		−43					
		Copper						r_2_ = 40 mm		2.39		−34					
**Polytetraflouroethene (PTFE)** [77,78,176]	PTFE	Copper	Copper	2.2	0.0009	25 × 50	127	Flat		3.5	13.3	−21	−19.5	NA			WiMAX X-Band
								r_1_ =50 mm		3.7	13.4	−16	−17.5				
								r_2_ = 40 mm		3.9	13.4	−15	−18				
	PTFE	Conductive Textile	Conductive Textile	2.1	0.002	95 × 90	500	Flat		2.25		−26		NA			ISM Wearable ES
								r_1_ = 200 mm		2.16		−38					
								r_2_ = 150 mm		2.75		−13					
	PTFE	Conductive Textile	Conductive Textile	2.1	0.0002	95 × 90 offset of slot 28.6 and stub 25.0	500	Flat		2.5	BW 0.23	−15		NA			ISM Band
								r_1_ = 225 mm		2.25	No BW	Below −10					
								r_2_ = 150 mm		2.25	BW 0.05	−11					
**Rogers PTFE Laminates** [222]	RO4003C	Copper	Copper	3.38	0.0027	80 × 60	200	Flat		3.0	3.8	−17	−20	NA			WLAN
								r_1_ = 100 mm		2.90	3.7	−18	−19				WiMAX
								r_2_ = 70 mm		2.95	7.7	−17	−16				
	RT/duroid 5880	Copper	Copper	2.20	0.0009	39 × 39	50.8	Flat		2.40–2.48 BW		−18		NA			ISM Band
								r_1_ = 60 mm		2.361–2.534		−17.9					
								r_2_ = 40 mm		2.382–2.552		−17.7					
								r_3_ = 20 mm		2.367–2.543		17.3					
	RO3003	Aluminium Silicon Nitride	NA	3.0	0.0010	-	128	Flat		7.24		−23.5		NA			UWB
								r_1_ = 40 mm		7.23		−18					
								r_2_ = 20 mm		7.25		−18.5					
**Liquid Crystal Polymers (LCP)** [10,81,149]	Rogers ULTRALAM 3850	Silver Nano Particles	Copper	2.9	0.0025	13.2 × 4.2	100	Flat		29		NA		6.4			Wireless Devices
								r_1_ = 6 mm		28				5			
								r_2_ = 4 mm		28				4.8			
	LCP	Copper	Copper	2.9	0.002	26 × 16	50	Flat		6.4	7.4	−22	−20	NA			UWB Wearable applications
								15 degree (min bent)		6.35	7.6	−20	−23				
								60 degree (max bent)		6.4	7.8	−21	−20				
	LCP	Copper	Copper	2.9	0.002	19 × 50	100	Flat		5.20		−24		NA			ISM
								30 degree (min bent)		5.20		−24					RFID
								60 degree (max bent)		5.25		−25					UWB

## Data Availability

Not applicable.

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
