# Peer review of "Bending Analysis of Polymer-Based Flexible Antennas for Wearable, General IoT Applications: A Review"

_polymers, 2021, doi:10.3390/polym13030357_

Round 1
Reviewer 1 Report
The introduction is clear and identifies the line of interest of this document, where the flexibility of polymeric materials is of paramount importance. Identifying in the literature extensive research on the subject. The review is to summarize comprehensively the bending capabilities of polymeric substrates for general IoT applications. The basic premise is to investigate the flexibility and bending capacity of polymeric materials, as well as their tendency to resist deformation.
It is an interesting subject and presents an adequate concentration of the state of art on the subject of interest and in chronological order. The tables presented can be clarified or designed in a better way, as they present a degree of difficulty in fully understanding them.
The state of the art is advisable to update and increase recent literature to define the relevance of the subject in greater depth.
The article is documented correctly, an adequate analysis of the various literature analyzed is presented. Identifying the properties of interest in the review.
The conclusions section can be improved and oriented to the most relevant characteristics with respect to the application(s) of interest.
Author Response
Dear Editors and Reviewers:
We would like to first thank the editor and reviewers for their feedback. The improved version of the manuscript was renamed to emphasize the main aim and stayed focused on the aim of developing, characterisation and modelling of the textile with recommendations for applications and bulk production for commercialisation of the research. The reviewers’ comments are specifically addressed below, with reference to the point in the manuscript into which feedback has been incorporated. Responds to the reviewer’s comments:
Reviewer 1: The introduction is clear and identifies the line of interest of this document, where the flexibility of polymeric materials is of paramount importance. Identifying in the literature extensive research on the subject. The review is to summarize comprehensively the bending capabilities of polymeric substrates for general IoT applications. The basic premise is to investigate the flexibility and bending capacity of polymeric materials, as well as their tendency to resist deformation.
It is an interesting subject and presents an adequate concentration of the state of art on the subject of interest and in chronological order. The tables presented can be clarified or designed in a better way, as they present a degree of difficulty in fully understanding them.
Response: We appreciate the reviewer comments on the theoretical section of the paper and agree with the reviewer that presentation of the tables is one of the important aspect of any articles. In this paper Table 4 titled “Comparison of General Requirements of Different Flexible Substrate for Antennas and Table 5 titled “Comparison of the Effects of Curvature on the Radiation Characteristics of Different Flexible Substrates for Mictrostrip Patch Antennas” are comprehensive and containing huge comparative data from various review articles, however we have tried our best to make them understandable and some modifications has been made in Table 4.
The state of the art is advisable to update and increase recent literature to define the relevance of the subject in greater depth.
The article is documented correctly, an adequate analysis of the various literature analyzed is presented. Identifying the properties of interest in the review.
The conclusions section can be improved and oriented to the most relevant characteristics with respect to the application(s) of interest.
Response: The conclusion part of the paper has been improved.
Reviewer 2 Report
This review report describes about polymer based flexible devices and their applications to IoT devices. The information looks useful and the manuscript is fundamentally well written. So, this reviewer recommend publication after addressing the following issues.
1.
Although this manuscript focuses on polymer (organic) based flexible devices. However, there are some examples of inorganic devices. For instance, the concept of integrated, thin, flexible devices using metal (S. Inoue et al., Appl. Phys. Lett. 88, 261910 (2006); T. W. Kim et al., Appl. Phys. Lett. 88, 121916 (2006)), mica or graphite sheet (Matsuki et al., Solid State Commun.136, 338 (2005)) and glass (Yalikun et al., Micromachines 7, 83 (2016)) were demonstrated. Authors should review such studies and indicate the advantages of using polymer.
Authors should provide more actual pictures of the devices for readers’ understanding. There are some small pictures in Fig. 3, but they are difficult to see the detail.
Fig.4 should have some more information. For instance, scale, thickness and materials should be added.
Author Response
Dear Editors and Reviewers:
We would like to first thank the editor and reviewers for their feedback. The improved version of the manuscript was renamed to emphasize the main aim and stayed focused on the aim of developing, characterisation and modelling of the textile with recommendations for applications and bulk production for commercialisation of the research. The reviewers’ comments are specifically addressed below, with reference to the point in the manuscript into which feedback has been incorporated. Responds to the reviewer’s comments:
Reviewer 2: This review report describes about polymer based flexible devices and their applications to IoT devices. The information looks useful and the manuscript is fundamentally well written. So, this reviewer recommend publication after addressing the following issues.
- Although this manuscript focuses on polymer (organic) based flexible devices. However, there are some examples of inorganic devices. For instance, the concept of integrated, thin, flexible devices using metal (S. Inoue et al., Appl. Phys. Lett. 88, 261910 (2006); T. W. Kim et al., Appl. Phys. Lett. 88, 121916 (2006)), mica or graphite sheet (Matsuki et al., Solid State Commun.136, 338 (2005)) and glass (Yalikun et al., Micromachines 7, 83 (2016)) were demonstrated. Authors should review such studies and indicate the advantages of using polymer.
Response: We appreciate the reviewer comments and suggestions. We agree with the reviewer and understanding the fact that some other inorganic materials have been used to provide flexibility for wearable applications in various literature such as graphite sheets, paper, transparent glass and fabrics. The use of all these materials as a substrate of FES or flexible antennas depends upon the required applications. Each of these materials has its own individual characteristics in terms of how efficiently they can be bent, twisted and/or crumpled. The principle drawbacks of most of these substrates are durability and washability which is overcome by polymer substrate and are being widely in used for wearable applications these days. Furthermore, To achieve the aforementioned characteristics for flexible antennas, conventional conductors and substrate materials such as metals and ceramics are not essentially appropriate. This is because these materials are usually rigid, costly, and lack flexibility and mechanical resilience. Therefore, we mainly focus on polymer substrates which covers the 90% of the flexible polymer based- industry to analyse physical, electrical, thermal and chemical properties of flexible polymer materials which have been used as a substrate to provide flexibility during the last few decades. For more reasons why we mainly focus on polymer substrates please see page 4 first three paragraphs of the paper.
- Authors should provide more actual pictures of the devices for readers’ understanding. There are some small pictures in Fig. 3, but they are difficult to see the detail.
Response: The suggested changes have been made
- Fig.4 should have some more information. For instance, scale, thickness and materials should be added.
Response: The Fig.4 represents a general structure of flexible antennas consist of a conductive layer and dielectric material backing and basic flexible microstrip patch antenna is a layer of thin conductive strip placed on the top of a flexible substrate. That’s why no specific scaling and parameters are shown in this figure. However, the parameters which include dimensions and thickness along with the substrate materials of antennas which are taken into review for this paper, are described in table 5.